# Dependency of the Charge–Discharge Rate on Lithium Reaction Distributions for a Commercial Lithium Coin Cell Visualized by Compton Scattering Imaging

**Kosuke Suzuki [1],*** , **Ryo Kanai [1], Naruki Tsuji [2], Hisao Yamashige [3], Yuki Orikasa [4], Yoshiharu Uchimoto [5], Yoshiharu Sakurai [2] and Hiroshi Sakurai [1]**

1   Graduate School of Science and Technology, Gunma University, Kiryu, Gunma 376-8515, Japan;
    t161d023@gunma-u.ac.jp (R.K.); sakuraih@gunma-u.ac.jp (H.S.)
2   Japan Synchrotron Radiation Research Institute (JASRI), SPring-8, Sayo, Hyogo 679-5198, Japan;
    ntsuji@spring8.or.jp (N.T.); sakurai@spring8.or.jp (Y.S.)
3   Material Platform Engineering Division, Toyota Motor Corporation, Toyota, Aichi 471-8572, Japan;
    hisao_yamashige@mail.toyota.co.jp
4   Department of Applied Chemistry, Ritsumeikan University, Kusatsu, Shiga 525-8577, Japan;
    orikasa@fc.ritsumei.ac.jp
5   Graduate School of Human and Environmental Studies, Kyoto University, Sakyo, Kyoto 606-8501, Japan;
    uchimoto.yoshiharu.2n@kyoto-u.ac.jp
*   Correspondence: kosuzuki@gunma-u.ac.jp; Tel.: +81-277-30-1714

**Abstract:** In this study, lithium reaction distributions, dependent on the charge–discharge rate, were non-destructively visualized for a commercial lithium-ion battery, using the Compton scattering imaging technique. By comparing lithium reaction distributions obtained at two different charge–discharge speeds, residual lithium ions were detected at the center of the negative electrode in a fully discharged state, at a relatively high-speed discharge rate. Moreover, we confirmed that inhomogeneous reactions were facilitated at a relatively high-speed charge–discharge rate, in both the negative and positive electrodes. A feature of our technique is that it can be applied to commercially used lithium-ion batteries, because it uses high-energy X-rays with high penetration power. Our technique thus opens a novel analyzing pathway for developing advanced batteries.

**Keywords:** lithium reaction distribution; in-operando measurement; Compton scattering imaging

## 1. Introduction

Although lithium-ion rechargeable batteries are already widely used in our daily life, demand for them is rapidly increasing, since the development of electric vehicles is attracting much attention all over the world. To further expand the use of electric vehicles, batteries require not only high capacities, but also high safety and long lifetimes. Moreover, people expect fast-charging batteries at a similar refueling speed as in the gasoline car. However, in a previous neutron diffraction study, it was reported that inhomogeneous reactions occur in the graphite-negative electrode at a high-speed charge–discharge rate [1]. These inhomogeneous reactions are related to the degradation of the battery performance; furthermore, they carry some great risks. Therefore, it is important to monitor lithium reactions directly in the batteries under in situ or in operando conditions. Although the neutron diffraction technique is a powerful tool for in operando measurement, it reveals the reaction mechanism through the change of the lattice parameter in the electrode materials.

Recently, we have been developing a method for directly monitoring lithium ions using a high-energy X-ray Compton scattering technique, called Compton scattering imaging [2–4]. This technique enables us to measure the reactions in the batteries under in situ and in operando conditions because it uses high-energy X-rays as an incident beam, which has high penetration power in the materials [5]. Moreover, a line-shape of the Compton scattered energy spectrum, called the Compton profile, is obtained from this incoherent scattering technique and depends on the element. Here, the Compton profile, $J(p_z)$, is shown in the following equation [6],

$$J(p_z) = \iint \rho(\mathbf{p}) dp_x dp_y \tag{1}$$

where $\rho(\mathbf{p})$ is the electron momentum density, $\mathbf{p} = (p_x, p_y, p_z)$ is the momentum, and $p_z$ is assumed to lie along the direction of the scattering vector. The $\rho(\mathbf{p})$ is expressed by the following [7,8],

$$\rho(\mathbf{p}) = \sum_j n_j \left| \int \Psi_j(\mathbf{r}) \exp(-i\mathbf{p} \cdot \mathbf{r}) d\mathbf{r} \right|^2 \tag{2}$$

where $\Psi_j(\mathbf{r})$ is the wavefunction of an electron in the *j*-state and $n_j$ is the electron occupation. The index j covers all constituent atoms and orbitals. Since the Compton profile is directly linked to the wavefunction of the electrons, its line-shape varies depending on each element. So far, we have demonstrated a method to quantify the lithium concentration from the change of the line-shape of the Compton profile (shape parameter analysis; *S*-parameter analysis) [3] and have successfully determined simultaneously lithium compositions at the positive and negative electrodes during a charge–discharge cycle [4]. A distinctive feature of our technique is that we can quantify the lithium in a working commercial lithium-ion battery using X-rays, although in many X-ray techniques this is difficult to quantify. Our quantification of the lithium can be done by using a very simple formula in the *S*-parameter analysis, as shown in the following section. Moreover, the Compton scattering technique can be applied to the materials composing the battery as well as the large-scale battery produced. In fact, we have studied the mechanism of electrode reactions in positive electrode materials, $Li_xMn_2O_4$, $Li_xCoO_2$, and $Li_xFePO_4$, using Compton profiles [9–11].

In this study, we visualize lithium reaction distributions at two different charge–discharge speeds by applying Compton scattering imaging to the commercial lithium-ion coin cell of VL2020 and discuss the change of the reactions depending on the charge–discharge speed.

## 2. Experimental Study

The Compton scattering experiment was performed with a 08 W high-energy X-ray beamline of SPring-8, Japan. The experimental configuration was the same as our previous in operando measurement of the lithium concentration [4]. The incident X-ray energy was 115.56 keV, and the scattering angle was fixed at 90°. The Compton scattered energy spectrum was measured by nine segments of a pure Ge solid-state detector. The probing volume of a sample was limited by an incident slit and a collimator slit, arranged between the sample and the detector. The size of the incident and collimator slits were 25 μm in height, 500 μm in width, and 500 μm in diameter. Hence, the observed region was 25 μm in height, 500 μm in width, and 500 μm in diameter. A sample was set on a movable stage along the *x*, *y* and *z* directions. The sample was a commercial coin-type lithium-ion rechargeable battery VL2020, made by Panasonic Corporation. This battery had a diameter of 20 mm and a thickness of 2 mm; it was composed of a $V_2O_5$ positive electrode (800 μm in thickness), LiAl alloy negative electrode (300 μm in thickness), olefin-based non-woven fabric separator, Al wire-netting spacer, and dimethoxyethane electrolyte. The state of charge (SOC) of the batteries was controlled by two charge–discharge rates: 1C and 0.2C. Here, 1C and 0.2C denote that the time between a fully discharged state and a fully charged state is 1 h and 5 h, respectively. The Compton scattering energy spectrum

was measured by changing the vertical and horizontal positions of the sample with respect to the incident X-rays mapping the lithium reaction distribution.

The Compton scattered energy spectra obtained were transformed to *S*-parameters. The *S*-parameter digitalizes the line-shape of a Compton scattered energy spectrum. As mentioned above, the line-shape of the Compton scattered energy spectrum changes through the elements, because the Compton profile reflects the electron momentum density distributions. The *S*-parameter is directly linked to the lithium concentration in the positive and negative electrodes. Here, the lithium momentum density is distributed at low-momentum regions; thus, the high *S*-parameter value corresponds to a high lithium concentration [12]. The *S*-parameter is defined through the following equations [3,4],

$$S = \frac{S_L}{S_H} = \frac{\int\limits_{-1}^{1} J(p_z)dp_z}{\int\limits_{-5}^{-1} J(p_z)dp_z + \int\limits_{1}^{5} J(p_z)dp_z} \tag{3}$$

where $S_L$ and $S_H$ are the areas under the Compton profile covering the low-momentum and high-momentum regions; parameters of $\pm 1$ and $\pm 5$ are the ranges within the low-momentum and high-momentum regions, respectively. In this study, analyses of the lithium reaction distributions and lithium concentrations, dependent on the charge–discharge rate, were conducted through the *S*-parameters.

## 3. Results and Discussion

Figure 1a shows the entire internal structure of the sample coin cell through the *S*-parameters. In this figure, the region of vertical position z < −0.1 mm and z > 1.4 mm corresponds to the battery's outer stainless-steel case (SUS); the region around z = 0 mm corresponds to the spacer; the region 0.1 < z < 0.35 mm corresponds to the LiAl negative electrode; the region 0.4 < z < 0.55 mm corresponds to the separator, and 0.6 < z < 1.3 mm corresponds to the $V_2O_5$ positive electrode. To study the details of the lithium reaction, we measured the Compton scattered energy spectrum precisely at the region of 0 < z < 0.7 mm. Figure 1b shows the variation of the *S*-parameters in the full-discharged state (SOC0) and full-charged state (SOC100) when the battery is charged and discharged by 0.2C. By charging the battery, the *S*-parameters increase by about 1.7% at the negative electrode; on the other hand, they decrease by about 1.8% at the positive electrode. Moreover, the separator position shifts toward the positive electrode direction by charging the battery. This means that the lattice volume of the negative electrode material expanded through the insertion of lithium ions. From the above, the *S*-parameter allows us to study lithium reaction distributions in the coin cell.

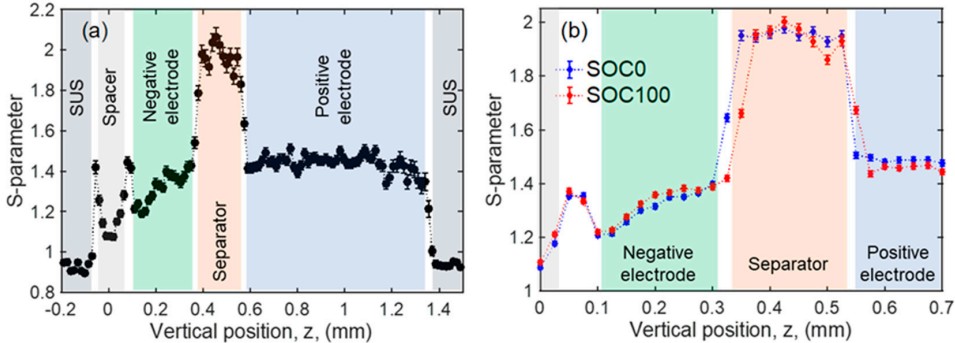

**Figure 1.** Internal structure of the VL2020 coin cell observed by the *S*-parameters. (**a**) Components of the coin cell are distinguished by the value of *S*-parameters. (**b**) Change of *S*-parameters at positive and negative electrodes with the charge–discharge cycle displayed. The fully charged state (state of charge (SOC) 100) and fully discharged state (SOC0) are shown in red and blue circles, respectively. The background color shows the regions of the corresponding components of the coin cell.

The lithium reaction distribution was obtained from two different charge–discharge rates of 0.2C and 1C, as shown in Figure 2. These distributions were obtained by scanning with the incident X-rays. The observed region on the coin cell is from the end of the negative electrode to the middle of the positive electrode, which is shown as a red square in Figure 2a. The size of the total mapping region is 0.7 mm in height and 10 mm in width.

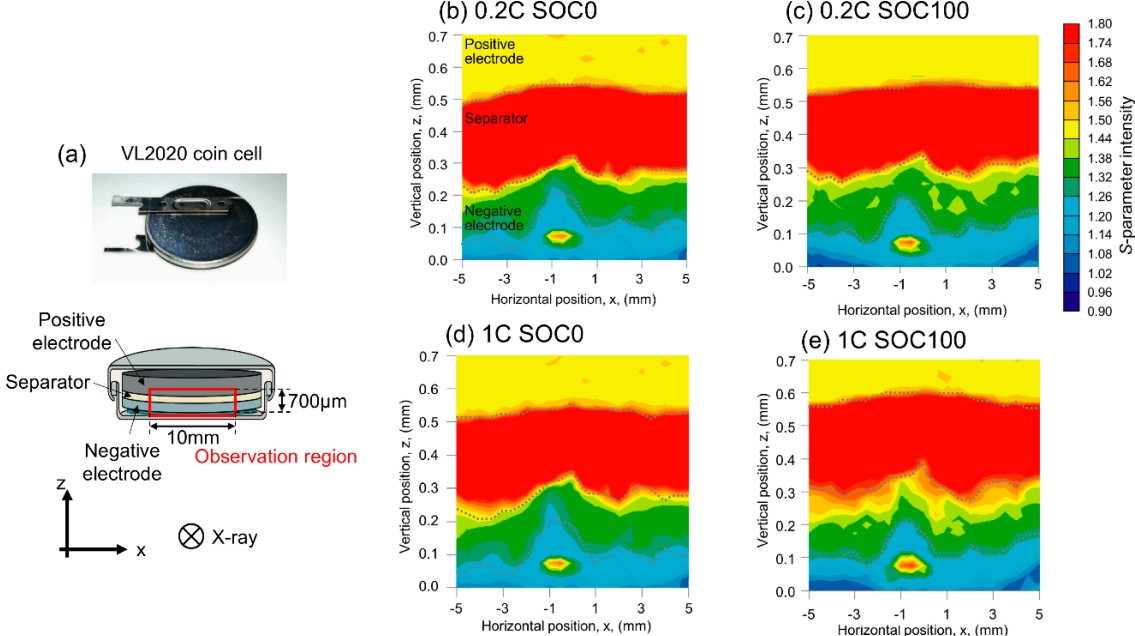

**Figure 2.** Lithium reaction distributions. (**a**) Picture of the sample and observation region of the lithium reaction distribution. (**b**) Lithium reaction distribution of the fully discharged state obtained at the 0.2C rate. (**c**) Lithium reaction distribution of the fully charged state obtained at the 0.2C rate. (**d**) Lithium reaction distribution of the fully discharged state obtained at the 1C rate. (**e**) Lithium reaction distribution of the fully charged state obtained at the 1C rate. The color corresponds to the *S*-parameter intensity in these reaction maps.

Figure 2b,c correspond to SOC0 and SOC100 of lithium reaction distributions at the 0.2C rate, and Figure 2d,e correspond to SOC0 and SOC100 of lithium reaction distributions at the 1C rate. In Figure 2b,d, the regions of high *S*-parameters are manifested at the region of the positive electrode; on the other hand, the regions of high *S*-parameters are manifested at the surface of the negative electrode in Figure 2c,e. It is observed that the separator position moves toward the positive electrode when the battery is charged. Although this trend is the same at the 0.2C and 1C rates, the amount of movement of the separator at the 1C rate is larger than that at the 0.2C rate. On the other hand, by discharging the battery, the separator position returns to the initial position. This movement of the separator position corresponds to the volume expansion or contraction of the electrode materials with moving lithium ions.

To exhibit the difference of lithium reaction distributions between 0.2C and 1C rates, we subtracted the lithium reaction distribution obtained at the 1C rate from that obtained at the 0.2C rate. Here, the separator position in the lithium reaction distribution obtained at the 0.2C rate was corrected to agree with the separator position obtained at the 1C rate following the same procedure as previously reported [4]. Figure 3a,b correspond to the change of the distributions in the SOC100 and SOC0, respectively. In Figure 3a, relatively vigorous lithium reactions occur at the surface of the negative electrode at the 1C rate. This trend of the lithium reactions occurring at the surface is the same as that reported in our previous study [4]. On the other hand, interestingly, when the battery was discharged at the 1C rate, as shown in Figure 3b, high *S*-parameter values appeared around the horizontal position of x = −1 mm in the negative electrode. This means that the lithium ions remain in the negative

electrode in the case of the 1C rate, although they move to the positive electrode in the fully discharged state of SOC0. These results were obtained by measuring non-destructively a commercial lithium-ion battery. Recently, the use of lithium metal as a negative electrode material has been considered to result in high-performance batteries, as the lithium metal has the highest theoretical capacity, 3861 mAh/g, among general negative electrode materials [13–15]. However, there is a problem that lithium dendrite occurs at the surface of the negative electrode. In our Compton scattering imaging system, the Ni refractive lens can be used and enables us to focus the vertical direction of the incident X-ray to 9 μm [16]. The spatial resolution on our measurement system reaches 9 μm in height, 50 μm in width and 50 μm in depth by combining the lens and the specially designed collimator slit system. Therefore, our technique can be exploited to monitor the generation of lithium dendrite non-destructively.

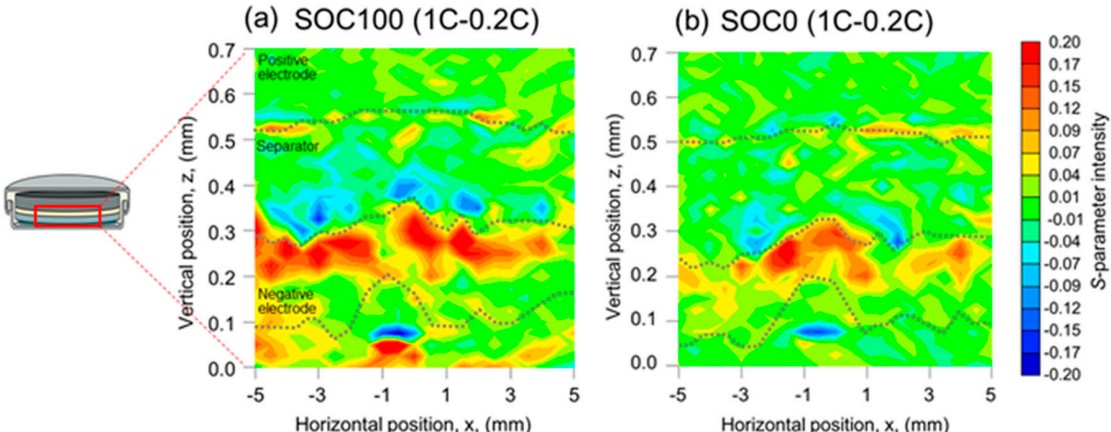

**Figure 3.** Change of lithium reaction distribution depending on the charge–discharge rate, for 1C and 0.2C rates. (**a**) At a fully charged state (SOC100). (**b**) At a fully discharged state (SOC0).

In Figure 4, an average of the *S*-parameters is shown, which is obtained from Figure 2 for every horizontal position, x, in the coin cell at the positive and negative electrodes; Figure 4a–d correspond to the positive electrode at SOC0, the negative electrode at SOC0, the positive electrode at SOC100, and the negative electrode at SOC100, respectively. The average values and deviations of the *S*-parameters, obtained from Figure 4, are summarized in Table 1. In Figure 4 and Table 1, the *S*-parameters in the positive electrode are almost constant, regardless of charge–discharge speed. In contrast, the *S*-parameters in the negative electrode exhibit some variation. Deviation of the *S*-parameters at the 1C rate is larger than that at the 0.2C rate, and the value becomes almost two times higher. To discuss the inhomogeneity of the lithium distributions, the *S*-parameters were converted to lithium composition by using calibration curves between the *S*-parameter and lithium composition of positive and negative electrodes. The following calibration curves were used: $x_p = 32.476 \times S_p - 47.186$ for the positive electrode and $x_n = 5.361 \times S_n - 6.967$ for the negative electrode; $x_p$ and $x_n$ show the lithium compositions of $Li_xV_2O_5$ and $Li_xAl$, and $S_p$ and $S_n$ show the averaged *S*-parameters per horizontal position. In Figure 5, the lithium composition for every horizontal position, x, of the battery is shown; Figure 5a–d correspond to the positive electrode at SOC0, the negative electrode at SOC0, the positive electrode at SOC100, and the negative electrode at SOC100, respectively. The average values and deviations for the lithium composition are summarized in Table 2. In Figure 5 and Table 2, we confirm that the deviation of the lithium composition obtained at the 1C rate is larger than that at the 0.2C rate. Therefore, inhomogeneous reactions are promoted during high charge–discharge speed.

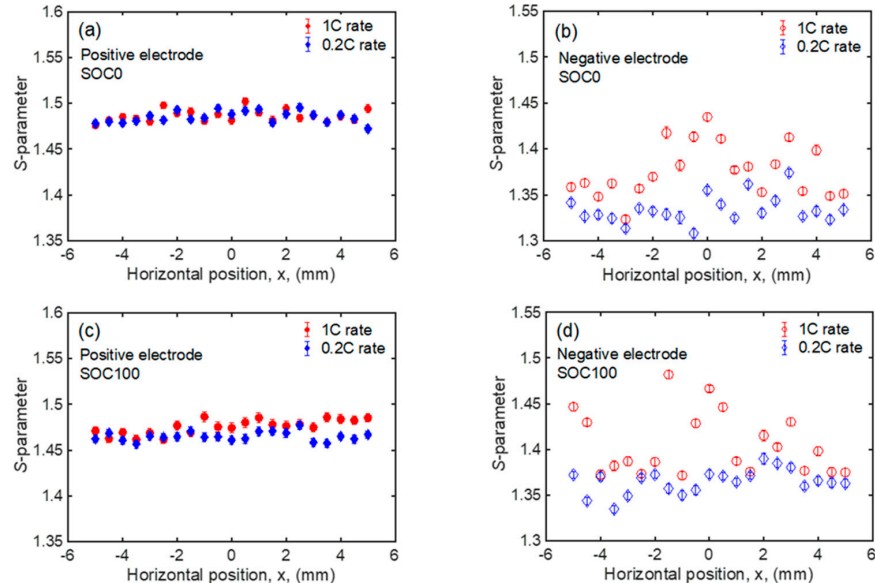

**Figure 4.** (**a**) *S*-parameters for each position at the positive electrode in SOC0. (**b**) *S*-parameters for each position at the negative electrode in SOC0. (**c**) *S*-parameters for each position at the positive electrode in SOC100. (**d**) *S*-parameters for each position at the negative electrode in SOC100. Blue and red symbols correspond to the results obtained at the 0.2C and 1C rates, respectively.

**Table 1.** Average values and deviations of *S*-parameters between the 1C and 0.2C rates.

| | | Positive Electrode | | Negative Electrode | |
|---|---|---|---|---|---|
| | | **SOC0** | **SOC100** | **SOC0** | **SOC100** |
| 1C rate | Average | $1.486 \pm 0.001$ | $1.476 \pm 0.001$ | $1.376 \pm 0.001$ | $1.405 \pm 0.001$ |
| | Deviation | 0.006 | 0.008 | 0.028 | 0.033 |
| 0.2C rate | Average | $1.485 \pm 0.001$ | $1.465 \pm 0.001$ | $1.334 \pm 0.001$ | $1.365 \pm 0.001$ |
| | Deviation | 0.006 | 0.005 | 0.015 | 0.013 |

**Figure 5.** Lithium compositions of each position at (**a**) the positive electrode in SOC0, (**b**) the negative electrode in SOC0, (**c**) the positive electrode in SOC100, (**d**) the negative electrode in SOC100. Blue and red symbols correspond to the results obtained from the 0.2C and 1C rates, respectively.

**Table 2.** Average values and deviations of the lithium composition for the 1C and 0.2C rates.

| | | Positive Electrode | | Negative Electrode | |
|---|---|---|---|---|---|
| | | SOC0 | SOC100 | SOC0 | SOC100 |
| 1C rate | Average | $1.088 \pm 0.028$ | $0.738 \pm 0.035$ | $0.412 \pm 0.005$ | $0.566 \pm 0.005$ |
| | Deviation | 0.210 | 0.252 | 0.151 | 0.179 |
| 0.2C rate | Average | $1.040 \pm 0.028$ | $0.387 \pm 0.032$ | $0.184 \pm 0.005$ | $0.349 \pm 0.005$ |
| | Deviation | 0.198 | 0.158 | 0.080 | 0.070 |

## 4. Conclusions

We applied the Compton scattering imaging technique for a commercial lithium coin cell. The motivation of this study was to deduce the charge–discharge rate dependency of the lithium reaction distribution by directly monitoring lithium ions. This is directly linked with the safety and longevity of batteries. In this study, we observed the residual lithium ions at the center of negative electrodes in a fully discharged state at a relatively high-speed discharge rate; the inhomogeneous reaction was facilitated at a relatively high-speed charge–discharge rate, in both the negative and positive electrodes. Compton scattering imaging can also be applied to commercialized large-scale lithium-ion batteries and enable in situ and in operando measurements of the battery. Therefore, the results are directly linked to the development of battery products.

**Author Contributions:** Conceptualization, K.S., Y.S., and H.S. with suggestions from H.Y., Y.O., and Y.U.; Compton Scattering Experiment and Data Analysis, K.S., R.K., N.T., Y.S., and H.S.; Writing-Original Draft Preparation, K.S.; Writing-Review, all co-authors; Writing-Final Editing, K.S., N.T., Y.S., and H.S.; Funding Acquisition, K.S. and Y.S.

**Funding:** This project was funded by [Japan Science and Technology Agency] and [MEXT KAKENHI] grant number [15K17873].

**Acknowledgments:** We thank M. Itou for technical support of the Compton scattering experiment. Compton scattering experiments were performed with the approval of JASRI [Proposal Nos. 2017A1123, 2017B1360 and 2018A1320].

**Conflicts of Interest:** The authors declare no conflict of interest.

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
