# Peer review of "Dependency of the Charge–Discharge Rate on Lithium Reaction Distributions for a Commercial Lithium Coin Cell Visualized by Compton Scattering Imaging"

_condensedmatter, doi:10.3390/condmat3030027_

Round 1

Reviewer 1 Report

This is a very good paper that shows how the microscopic structure of lithium batteries changes as a function of discharge rate. It presents a new technique that can be applied to ordinary samples, to gain critical information for diagnosing and improving the performance of battery products. The authors mention that this technique could be used to monitor the formation of lithium dendrites. This claim needs some further clarification, since one would expect that the dendrites must be larger than the spatial resolution resolution of the technique for them to be observed. A comment about the limits of the spatial resolution of this technique would be helpful for the reader in this regard.

Minor edits:

Line 68: Replace "proving" with "probing"

Line 107: Replace "allow" with "allows"

Line 179: Replace "batteries" with "battery"

Author Response

Reply to Reviewer Report (Reviewer 1):

Thank you very much for the constructive comments. All questions have now been addressed in point-by-point detail and appropriate revisions have been made in the manuscript.

Comments)

This is a very good paper that show how the microscopic structure of lithium batteries changes as a function of discharge rate. It presents a new technique that can be applied to ordinary samples, to gain critical information for diagnosing and improving the performance of battery products. The authors mention that this technique could be used to monitor the formation of lithium dendrites. This claim needs some further clarification, since one would expect that the dendrites must be larger than the spatial resolution of the technique for them to be observed. A comment about the limits of the spatial resolution of this technique would be helpful for the reader in this regard.

REPLY: We are grateful to the reviewer for carefully reading our manuscript and for funding our paper interesting. The part describing the spatial resolution of our Compton scattering imaging system has been added as follows and added new reference:  

 << In our Compton scattering imaging system, the Ni refractive lens can be used and enables us to focus the vertical direction of the incident X-ray to 9 mm [16]. The spatial resolution on our measurement system reaches 9 mm in height, 50 mm in width and 50 mm in depth by combining the lens and the specially designed collimator slit system. Therefore, our technique can be exploited to monitor the generation of lithium dendrite nondestructively. >>

Comment)

Minor edits:

Line 68: Replace “proving” with “probing”

Line 107: Replace “allow” with “allows”

Line 179: Replace “batteries” with “battery”

REPLY: We revised all points which pointed out to the reviewer. We performed the English proofreading of the article.

Reviewer 2 Report

The manuscript entitled "Dependency of Charge-Discharge Rate on Lithium
Reaction Distributions for a Commercial Lithium Coin Cell Visualized by Compton Scattering Imaging" reports a interesting and innovative work of impact in the lithium-ion battery field.
Before to accept the present manuscript the authors should be revise the Engilsh language and focus the introduction section on the the state of the art and as well as innovation of this work .

Author Response

Reply to Reviewer Report (Reviewer 2):

Thank you very much for the constructive comments. All questions have now been addressed in point-by-point detail and appropriate revisions have been made in the manuscript.

Comments)

The manuscript entitled “Dependency of Charge-Discharge Rate on Lithium Reaction Distributions for a Commercial Lithium Coin Cell Visualized by Compton Scattering Imaging” reports an interesting and innovative work of impact in the lithium-ion battery field.

Before to accept the present manuscript the authors should be revise the English language and focus the introduction section on the state of the art and as well as innovation of this work.

REPLY: We are grateful to the reviewer for carefully reading our manuscript and for good suggestion. The part describing the state of the art of our work has been added as follows and we performed the English proofreading of the article.: 

 << A distinctive feature of our technique is that we can quantitate the lithium in a working commercial lithium-ion battery using X-rays although in many X-ray techniques this is difficult to quantitate. Our quantitation of the lithium can be done by using a very simple formula in the S-parameter analysis, as shown in the following section. Moreover, the Compton scattering technique can be applied to the materials composing the battery as well as the produced large-scale battery. >>